# Study on the Motion Characteristics of Particles Transported by a Horizontal Pipeline in Heterogeneous Flow

**Jiayi Wang [1], Yitian Li [1,\*], Zhiqiang Lai [2,3], Lianjun Zhao [2,3] and Zhongmei Wang [2,3]**

1   State Key Laboratory of Water Resources and Hydropower Engineering Science, Wuhan University, Wuhan 430072, China
2   Yellow River Institute of Hydraulic Research, Yellow River Conservancy Commission, Zhengzhou 450003, China
3   Key Laboratory of Lower Yellow River Channel and Estuary Regulation, Ministry of Water Resources, Zhengzhou 450003, China
\*   Correspondence: hnjpsj@126.com

**Abstract:** The worldwide problem of reservoir sedimentation has perplexed the water conservancy industry. The problem of reservoir sedimentation is particularly serious in sandy rivers in China and directly affects the normal function of reservoirs. Due to its effect on the economy and environmental protection, the self-priming pipeline dredging and sediment discharge technology has broad application prospects. Nevertheless, there are pressing problems in the transportation of slurry particles in the pipeline system of this new technology. The purpose of this study is to use physical model tests to analyze the influence of the sediment transport rate and pipeline velocity on the motion state of particles (aggregation transport, jump transport, and suspension transport) when a heterogeneous flow with different particle sizes is transported in the pipeline. The results indicate that under the same pipeline velocity and sediment transport rate, the thickness of the static particle accumulation layer decreases with the increase in particle size in the state of aggregation and transportation, and the smaller the particle size, the greater the particle movement speed in the case of aggregation and suspension transportation. During jump transportation, the velocity of particles above the critical inflection point Y' increases with the decrease in particle size. The opposite is found below the critical inflection point Y'. At the same particle size and sediment transport rate, when the pipeline velocity increases, the particle transport transits from aggregation transport to jump transport and then to suspension transport. The larger the pipeline velocity, the greater the overall movement speed of particles. When gathering and conveying, if the pipeline flow rate increases by 1.5, the maximum movement speed of particles increases by 3.3. The curvature of the vertical velocity curve of the particles during jump transportation is not affected by the pipeline velocity. The particle velocity at the highest point increases with the increase in the pipeline velocity. During suspension transportation, the difference between the maximum and minimum vertical particle distribution velocities is exponentially related to the pipeline velocity. At the same pipe velocity and particle size, the overall particle velocity decreases with the increase in sediment transport rate.

**Keywords:** self-priming pipe; pipeline velocity; pipeline sand drainage; particle motion; sediment transport rate

## 1. Introduction

Reservoirs are a useful way for human beings to transform nature and use it to serve humankind. China has the largest number of reservoirs in the world. These reservoirs guarantee that China can feed 22% of the global population with only 6% of the world's renewable water resources. In addition, reservoirs also play a key role in maintaining the regional ecological balance, ensuring the water supply and utilization of biological resources, and reducing flood disasters. However, after the completion of reservoir construction

and water storage, the reservoir water level rises, the flow velocity in the reservoir area slows down, and the sediment transport capacity decreases, resulting in a large amount of sediment deposition in the reservoir area [1]. Sediment deposition accompanies reservoirs, and many reservoirs in the world are troubled by the problem of sedimentation. According to statistics, the annual sedimentation rate of global reservoirs is 0.5–1.0%, while that of Chinese reservoirs is 1.0–2.0%. A large amount of sediment deposition not only directly reduces the storage capacity and accelerates the loss of reservoir life, but also seriously hinders the normal operation of the water conservancy project. Sediment deposition has become a country-wide concern. At present, new dam sites available for reservoir construction are almost exhausted, and increasingly strong requirements for restoring the function of the reservoir through dredging have been proposed.

The research and practice of reservoir capacity restoration technology at home and abroad have undergone several centuries of development [2–5]. During this period, researchers have cited many representative achievements that have played an important role in reducing reservoir sedimentation, prolonging reservoir service life, and improving economic benefits. Reservoir capacity restoration technology is mainly divided into two categories: hydraulic sand removal and mechanical desilting. Hydraulic desilting causes the silted sediment in the reservoir area to be discharged out of the reservoir through the water release and desilting structure through reservoir regulation, mainly including density flow desilting, empty reservoir desilting, and other methods. In assessing the desilting effect, the effective range of hydraulic desilting is mainly concentrated in front of the dam, and the water consumption in the desilting process is huge. Mechanical desilting is a direct, reliable, and widely applicable method to remove silted sediment in river reservoir areas that has been used worldwide, such as in the dredging of the Suez Canal and Panama Canal, the desilting of Monterey Bay [6], the Mississippi River, the San Benito River, Dubai's Port of Jebel Ali, China's Yangtze River, Yellow River, Songhua River, Liaohe River and Haihe River basins. This includes the dredging of the Paulanica Reservoir in Switzerland, Kasukabe Reservoir in Japan [7], Aswan High Dam Reservoir in Egypt [8], and China's Guanting Reservoir [9], Jinping secondary, Yanguo Gorge, Wangjiaya, Hunan, Chenxing, Shuicaozi, Shiyan, and other reservoirs. At present, mechanical desilting mainly includes transportation machinery dredging, dredger desilting, pipeline desilting, and other methods.

Among these, the self-priming pipeline desilting technology uses the head difference between the upstream and downstream of the reservoir for the power to discharge the silted sediment in the reservoir area through the pipeline. This process does not need additional power, the project operation investment is small, the dredging cost is low, and the pump factor is not considered when transporting the slurry. Therefore, the hydraulic transportation of the various sized particle impurities within the pipe diameter range is allowed, which has received great attention from engineering circles at home and abroad. Many tests and studies have been conducted in reservoirs with appropriate conditions [10]. In 2019, on-site desilting tests of self-priming pipe desilting technology were carried out in reservoirs in Western China. The water platform was 11 m long and f 5 m wide (Figure 1a). The non-dynamic design was adopted. A winch was set in front of and behind the platform, which was connected to ground anchors on both banks through steel wire rope to allow movement of the platform along the predetermined section. The diameter of the sand discharge pipe was 0.2 m, the length of the pipe was 250 m, and the maximum installation height of the pipe was 0.97 M. Steel pipes and PE hoses were alternately assembled to ensure the flexibility of the water pipe. In order to prevent the pipe from sinking into the water bottom, a pair of foam pontoons with a diameter of 0.5 m were installed on the pipe at an interval of 2 m (Figure 1b).

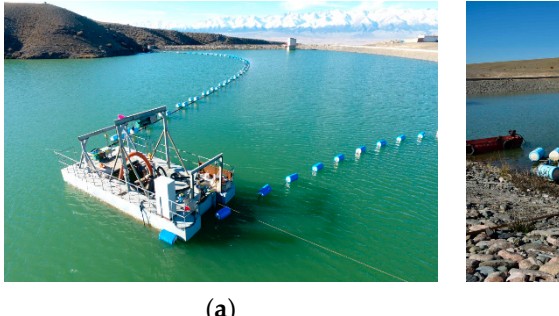 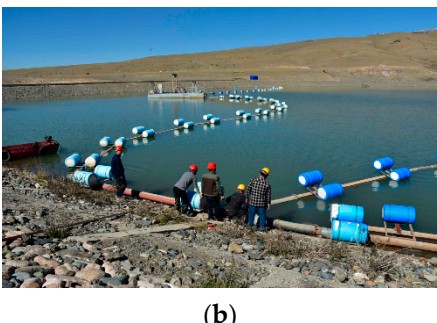

| (**a**) | (**b**) |

**Figure 1.** Field test of self-priming pipe sand removal technology. (**a**) Field test of reservoir desilting, (**b**) Desilting pipe.

Since the self-priming pipeline sand removal technology was not equipped with a power pump, pebbles, clay blocks, branches and other debris within the pipe diameter were allowed to enter the pipeline with the mud and the hard large particles entering the pipeline were not broken or pulverized [11]. Therefore, after pebbles and other coarse particles entered the pipeline, because their movement speed was significantly smaller than the flow velocity of hyperconcentrated sediment flow in the pipeline, they often could not be discharged out of the pipeline with the muddy water in time, which not only reduced the transportation efficiency of the pipeline sediment discharge system, but also easily accumulated in the pipeline and blocked the pipe. For example, at the dredging operation site, there was a water hammer oscillation of the pipeline system caused by particle blockage, with a maximum oscillation amplitude of 70 cm, which caused serious safety hazards to the project operation and also introduced new problems to the pipeline transportation technology [12]. Figure 2 shows the pebbles and coarse particles discharged from the pipe outlet. After field measurement, the particle size of the pebbles and coarse particles was shown to be between 2 and 60 mm.

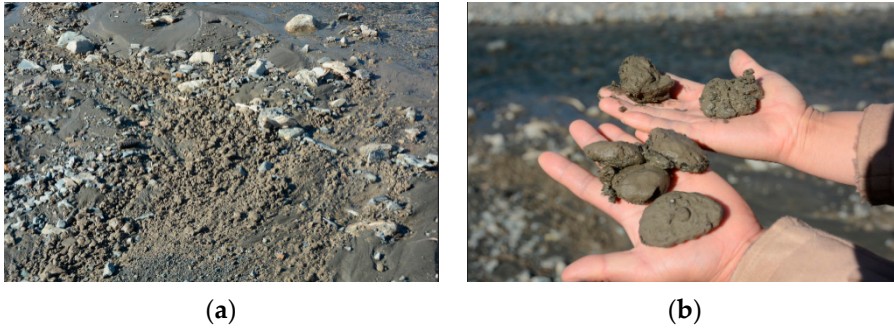

| (**a**) | (**b**) |

**Figure 2.** Sediment discharged from the test pipeline at the dredging site. (**a**) Sediment discharged from the pipeline, (**b**) Pebbles discharged from the pipeline.

Although experts and scholars have carried out much research on pipeline slurry and particle transportation, owing to the complexity of solid–liquid two-phase flow and engineering application conditions, many problems existing in practical engineering have not actually been solved, such as the motion characteristics of coarse particle groups in the pipeline and the vertical velocity distribution characteristics and calculation methods of coarse particle groups in the pipeline [13–15]. Research on the motion characteristics of particles and the vertical distribution of the velocity in horizontal pipelines has been carried out to study the influence of the law of particle size, pipe flow velocity, and sediment transport rate on the vertical distribution of particle group velocity under common working conditions, which will not only guide the design of artificial dredging equipment and prevent particle accumulation and siltation in pipelines but also provide a theoretical reference for the study of pipeline particle transport in other engineering fields [16–19].

The purpose of this study is to study the effects of particle size, pipeline velocity, and sediment transport rate on the vertical distribution characteristics of particle velocity in the pipeline under the three transport states of aggregation, jump, and suspension through generalized model tests. It provides a scientific basis for sand discharge by self-priming pipelines.

## 2. Horizontal Pipeline Particle Conveying Test Device

### 2.1. Model Design

In order to fully study the law of particle movement in pipeline water flow, a physical model test device for pipeline particle hydraulic transportation was designed and established (Figure 3). The physical test device was composed of an imported mixing tank and control system, a plexiglass pipe section, and a water return system.

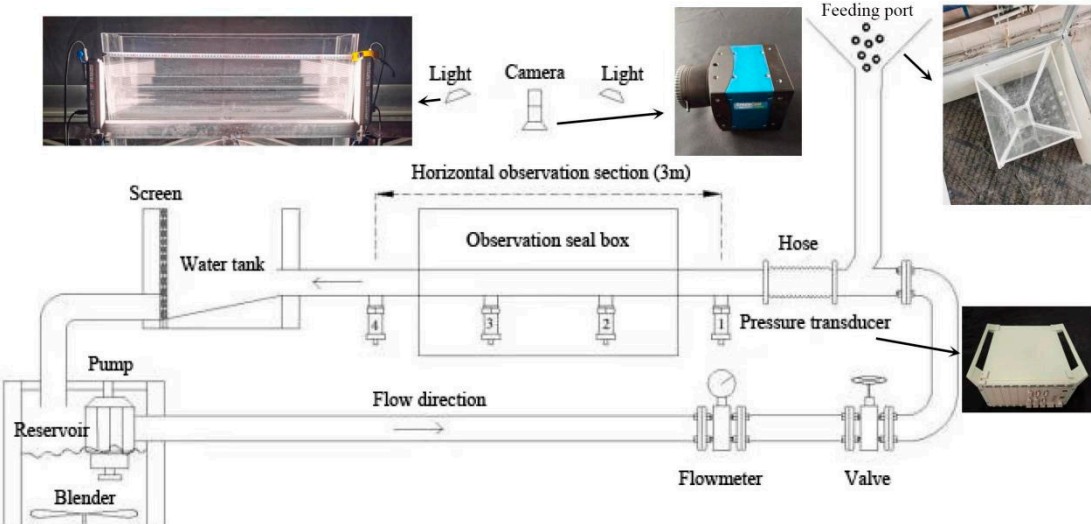

**Figure 3.** Solid model generalization diagram.

The diameter of the plexiglass pipeline was 0.1 m, and the total length was 23 m. Through the numerical variation amplitude of four pressure sensors uniformly laid along the lower part of the pipe wall, the middle 3 m was determined as the effective observation section. Pressure measurement points numbered 1–4 were located along the pipe wall at intervals of 1 m, the sampling frequency was 50 Hz, and the sampling time interval was 0.02 s. A particle feeding port was arranged in the front of the effective horizontal observation section, and the feeding port was located 2 m upstream of the effective observation section. The particles entered the pipeline through the feeding port and were filtered out by the filter in the backwater tank to prevent the particles from entering the mixing tank.

The pipeline fluid power was provided by the water pump, and the overflow was controlled by the electromagnetic flowmeter and valve to meet the fluid transmission speed required for the test. The particle motion state was photographed and recorded by using a high-speed camera, and the fill light further improved the clarity of the photographs. At the same time, a plexiglass observation sealing box filled with clean water was arranged outside the pipe. The sealing box was 0.8 m long, 0.1 m wide, and 0.5 m high, so as to eliminate distortions in the photographs caused by different refractive indexes as much as possible.

### 2.2. Model Materials and Instruments

In order to ensure the consistency of particle shape under different test conditions, special 35% zirconia grinding beads (round balls) were selected as test materials after comparing various model materials. The density of material was 2.65 kg/m$^3$, and the refractive index was 1.78. The difference between the physical and mechanical properties

of density, recovery coefficient, stiffness, and other parameters of natural pebble was no more than 3%, and the physical and mechanical parameters of each particle were basically the same. Particle sizes were 2, 4, 6, 8, and 10 mm. Figure 4 shows the photos of particles selected in this experiment.

The shape distribution of natural pebbles was random. If natural pebbles had been directly used in the test, we could not ensure that the particle shape was completely consistent under different test conditions. Therefore, after comparing various simulation materials, the special 35% zirconia grinding beads (balls) were finally selected as the test materials. The density of this material was 2.65 kg/m$^3$ and the refractive index was 1.78. The difference between the density, recovery coefficient, stiffness, and other physical and mechanical properties of natural pebbles was no more than 3%, and the physical and mechanical parameters of each particle were basically the same. Particle size D was 2, 4, 6, 8, and 10 mm, respectively. Figure 4 shows the photo of the particles selected in this test.

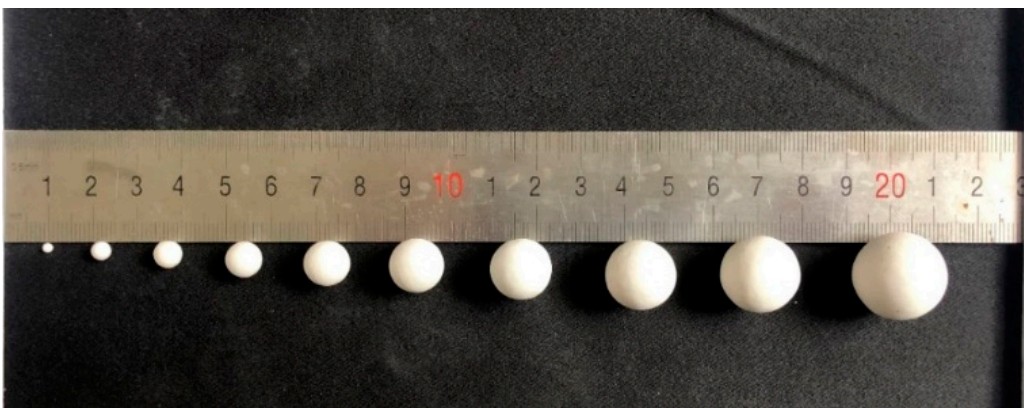

**Figure 4.** Diagram of 35% zirconia ceramic beads used in the test.

The instruments, equipment, and software used in the model test included the high-speed camera PTV measurement system, pipeline pulsation pressure sensor and DHDAS pressure acquisition system, electromagnetic flowmeter, frequency converter, electronic scale, mobile storage hard disk, camera, etc.

(1) The high-speed camera PTV measurement system includes a lighting system, high-speed camera, and PTV image processing system. The lighting system uses Altos AL-528 photographic fill light, which has the advantages of strong illumination, sensitive adjustability, wide lighting angle, and so on. The high-speed camera is one of the HSVISION SpeedCam series, with a frequency of 250 Hz, a time delay of 4 ms between two images, and 2048 × 2048 image pixels. The PTV image processing system uses the particle image underwater section measurement system optical flow technology software of Qingdao Co., Ltd. (Qingdao, China), to process the pictures taken by the high-speed camera to obtain the particle velocity distribution, motion morphology, motion trajectory, and so on, as shown in Figure 5. The PTV image processing system utilizes the classical cross-correlation algorithm to process the images from two consecutive frames in a grid, divide them into different query windows according to the grid settings, calculate the similarity of particle images in the query window, obtain the particle displacement in each query window through peak detection, and then obtain physical quantities such as velocity and acceleration.

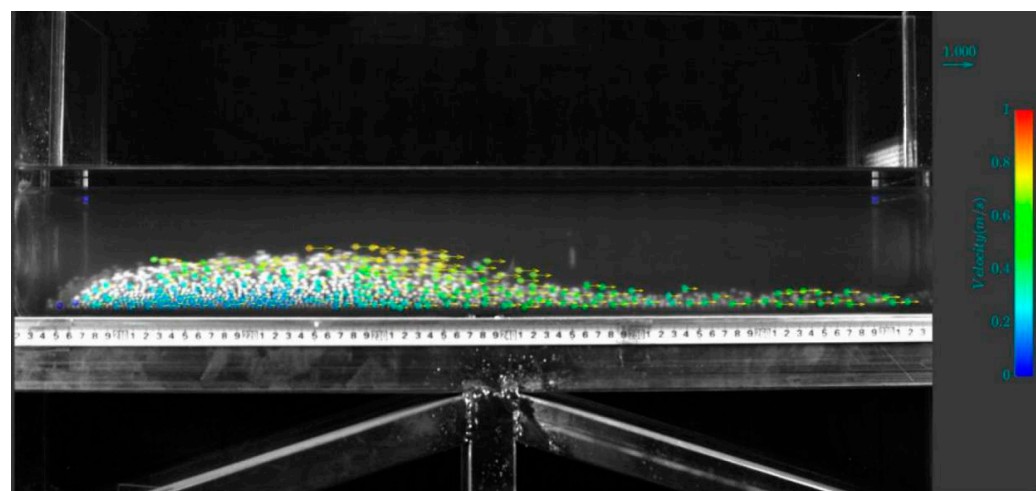

**Figure 5.** Operation interface of the PTV measurement system for pipeline fluid particle transportation.

(2) The pipeline pressure sensor and DHDAS pressure signal acquisition and analysis system use a sputter film pressure sensor to measure the variation law of pulsating pressure at the bottom of the pipeline. The working principle is that when the measured medium pressure acts directly on the metal film of the sensor, the film produces deformation proportional to the medium pressure, and the bridge outputs the electrical signal corresponding to this pressure. Because the diaphragm is made of high-performance elastic material, the sensor has strong overload resistance and high measurement accuracy. After the preliminary pressure measurement test, we found that the maximum pressure at the maximum flow rate of the pipeline was not more than 10 KPA. Therefore, we used 0~40 kPa range sensors to measure the pipeline pressure, and each sensor underwent a separate sensitivity check before installation. The DHDAS pressure signal acquisition and analysis system has 128 measurement channels, and the sampling frequency range is 10~128 kHz.

(3) Once powered on, the electromagnetic flowmeter and frequency converter directly read the current flow value through the display screen of the electromagnetic flowmeter, and the accuracy of the flowmeter can be checked before the test. During flow control, the valve cooperates with the frequency converter for flow control. The frequency adjustment range of this frequency converter is 0~50 Hz, and the accuracy is 0.001 Hz.

*2.3. Test Method*

Before the test, according to the sediment transport rate of particles and the test simulation time, the particles required for this test were weighed with an electronic scale, the pressure sensor connected, and three cameras and high-speed cameras were fixed at appropriate positions to capture and record the particle movement process in the pipeline. We opened the water valve and water pump and adjusted the flow to the level required by the test by observing the flowmeter; after the flow stabilized, we turned on the three cameras and a high-speed camera PTV to start recording. At the same time, we activated the DHDAS dynamic signal acquisition and analysis system to record pressure changes. Finally, particles were evenly added to the pipeline within the test simulation time. During the test, the start and end of particle feeding and the time when particles completely leave the pipeline were recorded.

*2.4. Test Group*

Here, physical experiments on particle movement in heterogeneous flow in a horizontal pipeline under the conditions of different particle sizes, transportation volumes and pipeline velocities were conducted. Particle size d was selected as 2, 4, 6, 8, and 10 mm, defined as $d_2, d_4, \ldots, d_{10}$, respectively. The conveying capacity was set to uniformly release 4.16, 12.5, and 20.8 kg of particles within 20 s, respectively, and the corresponding

sediment transport rate was 0.208, 0.625, and 1.04 kg/s, defined as $v^s{}_{0.208}$, $v^s{}_{0.625}$, and $v^s{}_{1.04}$, respectively. The pipeline velocity $v$ was set as 1.0, 1.5, 2.0, 2.5, and 3.0 m/s, defined as $v_{1.0}$, $v_{1.5}$, $v_{2.0}$, $v_{2.5}$ and $v_{3.0}$, respectively. The physical test conditions were named according to the particle size, the pipeline's flow velocity and the particle transport volume; for example, the test condition $d_2v_{1.5}v^s{}_{0.208}$ indicated that the particle size $d_2$ was 2 mm, the pipe flow velocity $v_{1.5}$ was 1.5 m/s and the particle conveying capacity was 4.16 kg particles conveyed uniformly every 20 s. The experimental temperature was 15 °C, the Reynolds number for all tests was $Re > 10^5$, and the water flow in each group was in the square resistance zone (Table 1).

The Froude number is calculated using the following equation:

$$Fr = (v^2/gL)^{0.5} \tag{1}$$

where $Fr$ is the Froude number; $v$ is the moving speed of the object; $g$ is the acceleration of gravity; $L$ is the characteristic length of the object.

**Table 1.** Test groups.

| Number | Particle Size (mm) | Particle Conveying Capacity (kg) | Particle Transport Rate (kg/s) | Pipeline Velocity (m/s) | Reynolds Number |
|---|---|---|---|---|---|
| 1 | 2/4/6/8/10 | 4.16 | 0.208 | 1.0 | 87,719 |
| 2 | 2/4/6/8/10 | 12.5 | 0.625 | 1.5 | 131,579 |
| 3 | 2/4/6/8/10 | 20.8 | 1.04 | 2.0 | 175,439 |
| 4 | 2/4/6/8/10 | 4.16 | 0.208 | 2.5 | 219,298 |
| 5 | 2/4/6/8/10 | 12.5 | 0.625 | 3.0 | 263,158 |
| 6 | 2/4/6/8/10 | 20.8 | 1.04 | 1.0 | 87,719 |
| 7 | 2/4/6/8/10 | 4.16 | 0.208 | 1.5 | 131,579 |
| 8 | 2/4/6/8/10 | 12.5 | 0.625 | 2.0 | 175,439 |
| 9 | 2/4/6/8/10 | 20.8 | 1.04 | 2.5 | 219,298 |
| 10 | 2/4/6/8/10 | 4.16 | 0.208 | 3.0 | 263,158 |
| 11 | 2/4/6/8/10 | 12.5 | 0.625 | 1.0 | 87,719 |
| 12 | 2/4/6/8/10 | 20.8 | 1.04 | 1.5 | 131,579 |
| 13 | 2/4/6/8/10 | 4.16 | 0.208 | 2.0 | 175,439 |
| 14 | 2/4/6/8/10 | 12.5 | 0.625 | 2.5 | 219,298 |
| 15 | 2/4/6/8/10 | 20.8 | 1.04 | 3.0 | 263,158 |

## 3. Results

### 3.1. Particle Velocity of Heterogeneous Flow in a Horizontal Pipeline

It is of key guiding significance for the parameter design of a pipeline transportation system to explore the movement state change law of particles in the pipeline [20–26]. The movement state of particles during pipeline hydraulic transportation changes with the changes in pipe flow velocity, particle size, sediment transport rate, and other factors. Based on the 75 groups of particle motion state diagrams of clear water transportation in horizontal pipelines from photographs taken by the high-speed-camera PTV observation system, the particle motion states under different working conditions were divided into three types: aggregation transportation, jump transportation and suspension transportation, as shown in Figure 6.

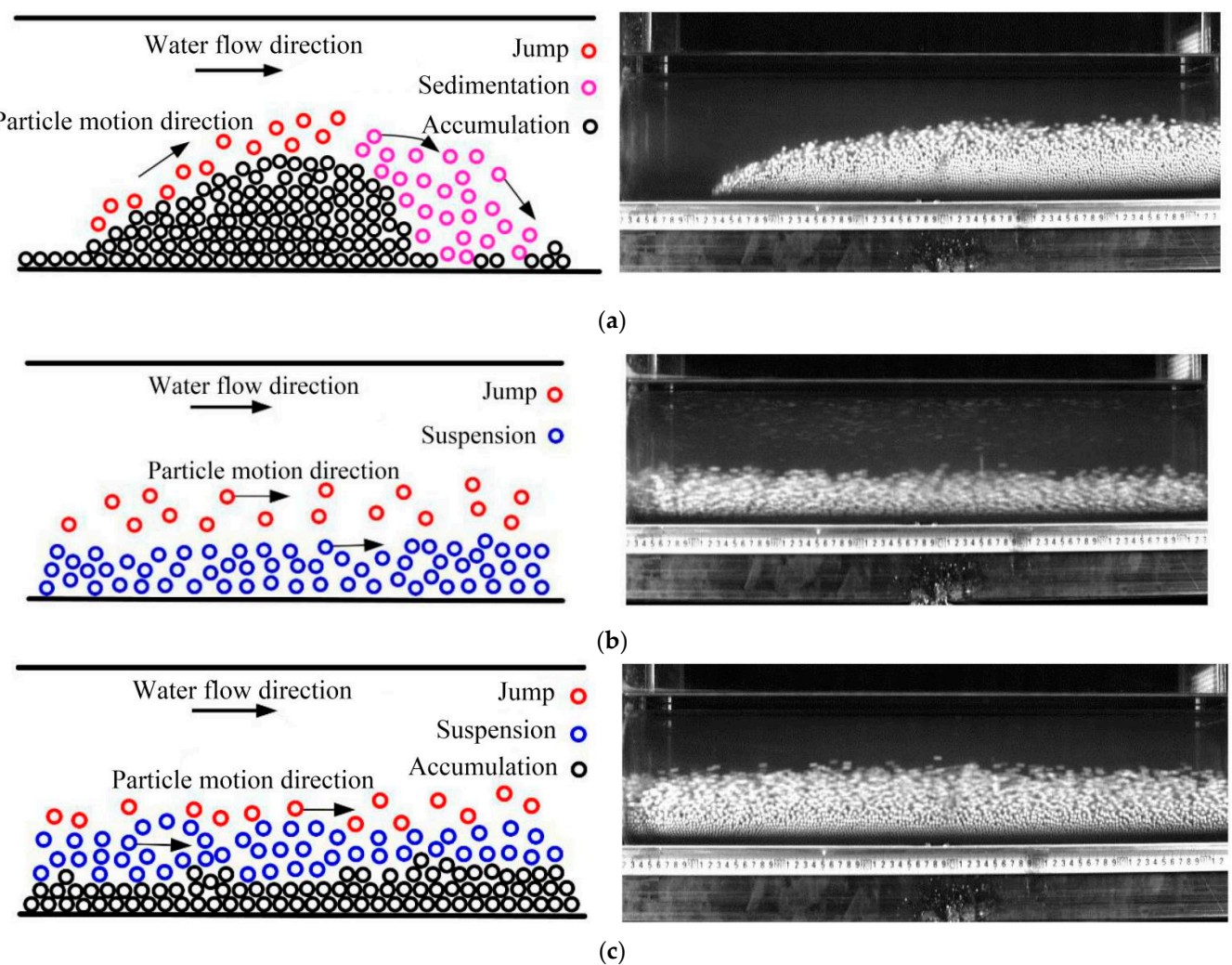

**Figure 6.** Motion state diagram of coarse particles in a pipeline. (**a**) Aggregate conveying; (**b**) Jump conveying; (**c**) Suspension conveying.

For aggregation and transportation, only the surface and upper particles in contact with the water flow have movement speed, and the movement speed of the bottom particles is basically zero. The whole particle cluster rolls and moves head to tail, which is similar to the sand wave movement law in river dynamics. In jump transportation, the movement of particles presents an obvious stratification phenomenon. The upper particles move at a high speed, with many tumbling and rotating movements. The lower particles move at a low speed, and the movement form is basically parallel displacement. In suspension transportation, the particles are dispersed in the pipeline water flow. When the particle size is small, the particles are mostly suspended in the water flow. When the particle size is large, the particles are close to the bottom of the pipeline, the particles move rapidly in the pipe, and the particles are in discrete contact with each other.

*3.2. Analysis on Influencing Factors of Vertical Distribution of Particle Velocity in Horizontal Pipeline*

According to the physical model test results, the vertical distribution characteristics of particle velocity in different motion states (aggregation transportation, jump transportation, suspension transportation) and the influence of pipeline velocity, particle size and sediment transport rate on the vertical distribution of particle velocity were analyzed.

Taking the bottom of the horizontal pipe as the coordinate zero point, the *X* axis represents the moving speed of particles along the water flow direction of the pipe (m/s),

and the $Y$ axis represents the distance (m) between particles and the bottom of the pipe. The moving speed of particles along the water flow direction of the pipe at the height of y is defined as $v(y)$. PTV image processing system was used to process the pictures taken by the high-speed camera to obtain the particle speed $v(y)$. The pipe diameter is $D$ (mm), the particle size is $d$ (mm), and the flow velocity in the pipeline is $V$ (m/s). Figure 7 shows the physical parameters of fluid particles in the pipeline.

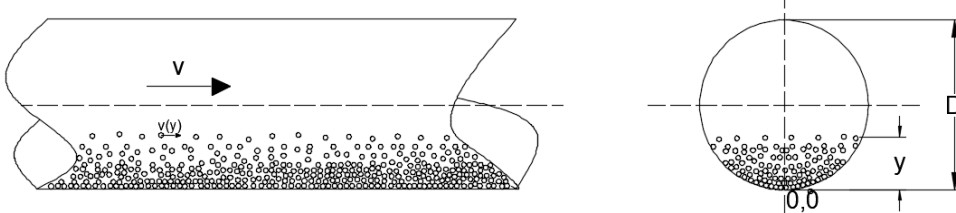

**Figure 7.** Schematic diagram of the physical parameters of fluid particle motion in a pipeline.

3.2.1. Influence of Particle Size and Vertical Distribution of Particle Velocity

Under the same pipeline flow rate and sediment transport rate, the thickness of the stationary particle accumulation layer decreased with an increase in the particle size, while the thickness of the accumulation layer decreased threefold with an increase in the particle size. Close to the center of the pipe, the larger the particle size, the smaller the velocity (Figure 8a). The ratio of deposit thickness to particle size is defined as the equivalent deposit thickness $\Delta H$. Analysis of the equivalent accumulation thickness by aggregation transportation $\Delta H$ and particle size d (Figure 8b) revealed that the $\Delta H \infty d^{-n}$ relationship could be tested and calibrated, where n = 1.508. Therefore, the accumulation thickness of aggregation transportation can be calculated by the equivalent accumulation thickness: $\Delta H = 33.89 d^{-1.508}$.

During saltation transportation, the particle velocity of the same section showed obvious stratification, and the critical inflection point $Y'$ of particle velocity stratification of different particle sizes was basically the same. Under the same pipeline velocity and sediment transport rate, the larger the particle size, the greater the movement speed. The movement speed of particles above the critical inflection point $Y'$ was negatively correlated with the particle size. The analysis showed that particles below the critical inflection point $Y'$ formed a tile layer with a certain thickness at the bottom of the pipeline, resulting in a reduction in the cross-sectional area of the pipeline and an increase in the velocity of the pipeline above the tile layer, so the phenomenon of velocity stratification appeared. The maximum particle velocity at the same section increased with a decrease in the particle size (Figure 8c). Under the pipeline flow rate and sediment transport rate, for a particle size lower than the critical inflection point $Y'$, the larger the particle size, the larger the particle speed, and a particle size higher than the critical inflection point $Y'$ is negatively correlated with the particle size.

During suspension transportation, the particle velocity of the same section increased linearly in the vertical direction. Under the same pipeline flow and sediment transport rate, the smaller the particle size, the greater the particle movement velocity. The influence of particle size on the vertical velocity distribution of particles decreased with an increase in particle size (Figure 8d).

It can be concluded that during aggregation transportation and suspension transportation, the smaller the particle size, the greater the particle motion speed. During saltation transportation, the movement speed of particles above the critical inflection point $Y'$ increases with a decrease in the particle size, but the opposite occurs below the critical inflection point $Y'$.

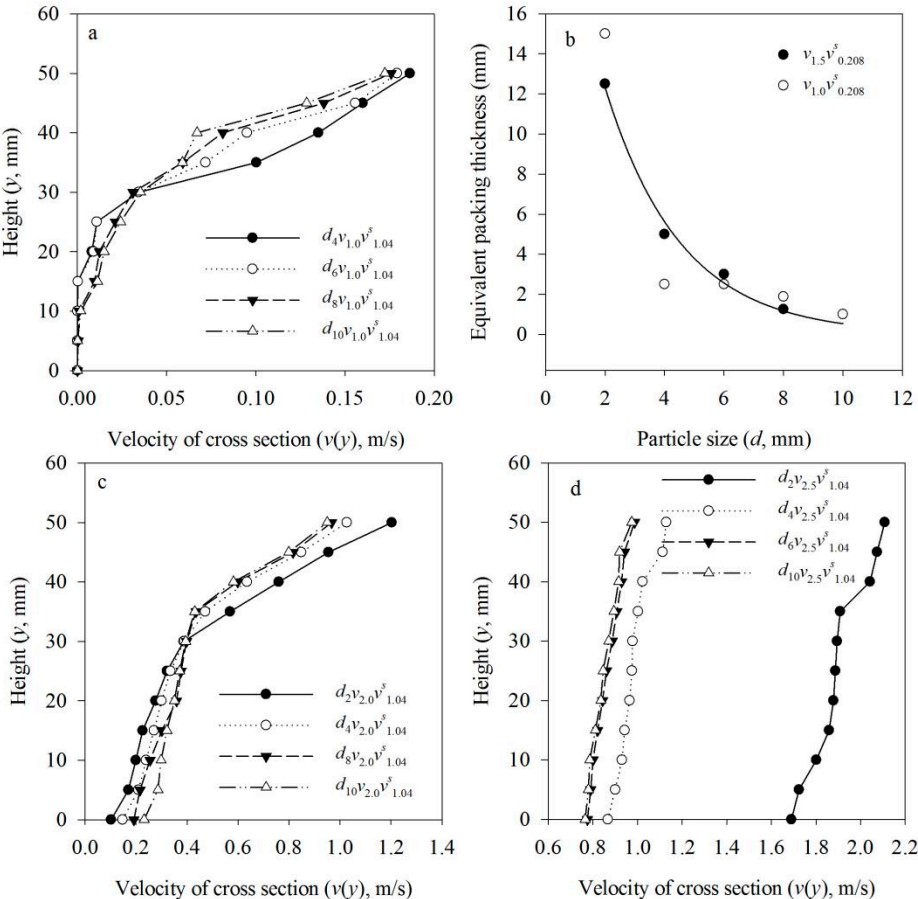

**Figure 8.** Particle size and vertical velocity distribution of particles. (**a**) the relationship between vertical velocity distribution and particle size in aggregate transportation; (**b**) aggregate transportation, equivalent packing thickness and particle size relationship; (**c**) the relationship between vertical velocity distribution and particle size in saltation transport; (**d**) the relationship between vertical velocity distribution and particle size in suspended transport.

### 3.2.2. Effects of Pipeline Velocity on the Vertical Velocity Distribution of the Particles

Under the same particle size and sediment transport rate, when the pipeline's flow rate increased, the particle transport shifted from aggregation transportation to saltation transportation, and then to suspension transportation. The greater the pipeline's flow rate, the greater the overall movement speed of particles (Figure 9). During aggregation transportation, the pipeline's flow rate increased 1.5 times and the maximum particle movement speed increased 3.3 times (Figure 10a). The curvature of the vertical velocity curve of particles during saltation transportation was not affected by the pipeline velocity, and the particle velocity at the highest point increased with an increase in pipeline velocity. During suspension transportation, the difference between the maximum and minimum values of the particles' vertical distribution velocity had an exponential relationship with pipeline velocity (Figure 10b).

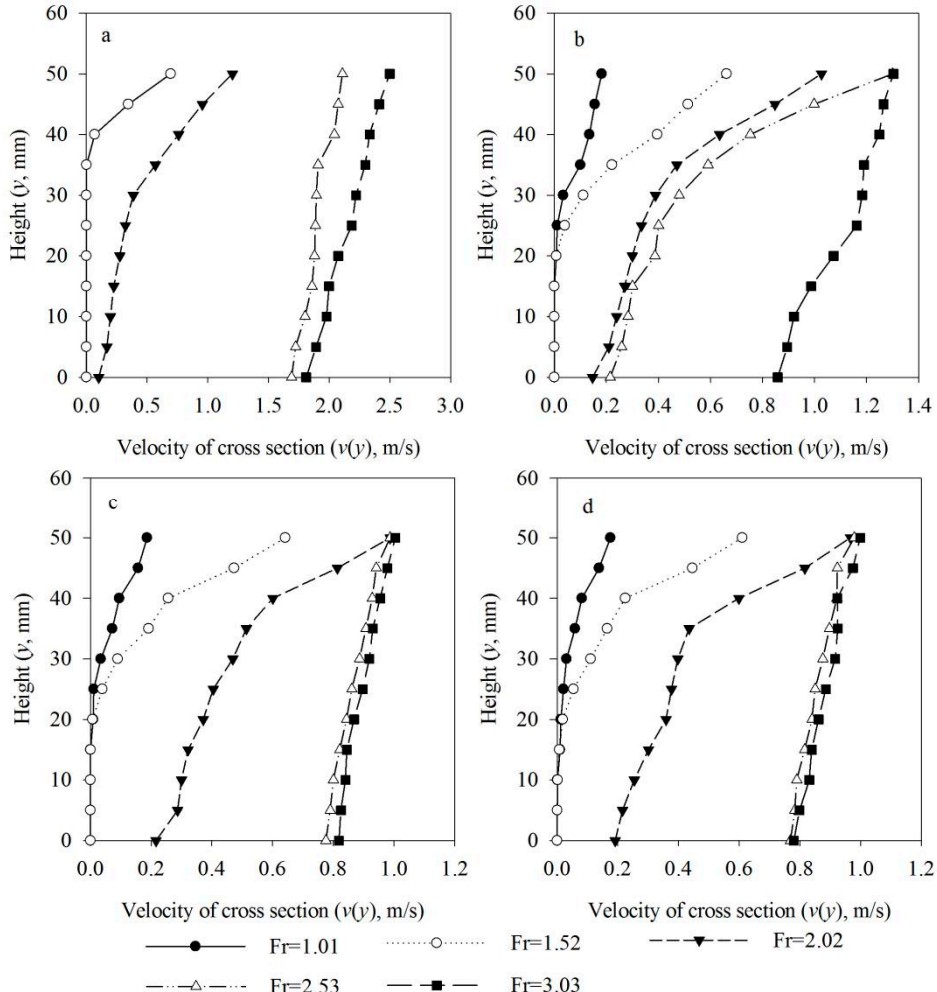

**Figure 9.** Relationship between vertical velocity distribution of particles and pipeline velocity. Fr, Froude number; $d$, particle size; $D$, pipe diameter; (**a**) $d/D = 0.02$, $v^s_{1.04}$; (**b**) $d/D = 0.04$, $v^s_{1.04}$; (**c**) $d/D = 0.06$, $v^s_{1.04}$; (**d**) $d/D = 0.1$, $v^s_{1.04}$.

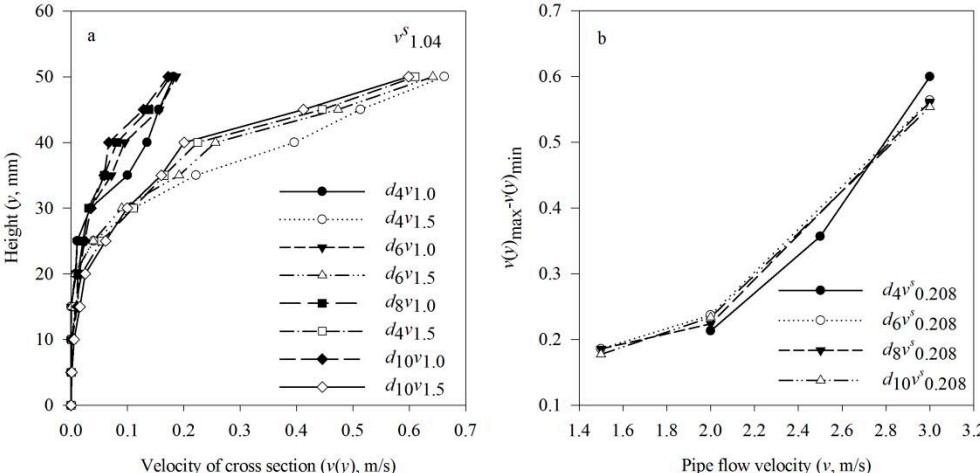

**Figure 10.** Relationship between vertical velocity distribution of particles and pipeline velocity. (**a**) gathering transportation; (**b**) suspension transportation; $v(y)_{max}$, maximum vertical distribution velocity of particles; $v(y)_{min}$, minimum vertical distribution velocity of particles.

### 3.2.3. Effect of Sediment Transport Rate on Vertical Distribution of Particle Velocity

During aggregation and transportation with the same particle size and pipe flow velocity, the larger the sediment transport rate, the smaller the overall movement speed of the particles, and the greater the thickness of the accumulation layer. This led to the smaller cross-sectional area of the overflow, the increase in the cross-sectional velocity, the sudden increase in the starting velocity of the particles in the upper layer, and the steepening of the vertical distribution curve of the particle movement velocity. During jump transport and suspension transport, the overall movement speed of the particles also decreased with the increase in the sediment transport rate (Figure 11).

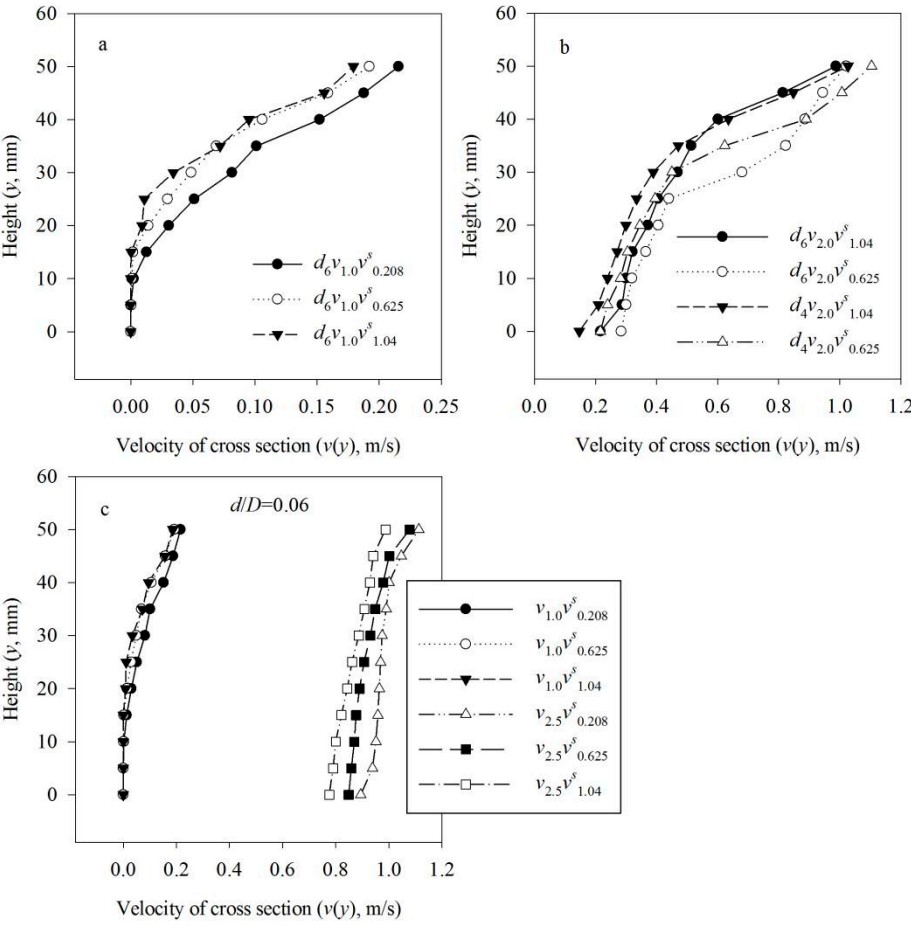

**Figure 11.** Relationship between particle vertical velocity distribution and sediment transport rate. (**a**) aggregate transportation; (**b**) saltation transportation; (**c**) suspension transportation.

## 4. Discussions

### 4.1. Horizontal Pipes Transport Particles

The most prominent characteristic of solid–liquid two-phase flow is that with different parameters such as the specific gravity of solid particles, pipe diameter, slurry temperature, slurry velocity, and so on, solid particles show different motion states. The larger the particles and the heavier the weight, the more complex the motion state. Scholars measure the macro motion characteristics of the fluid and solid through pipeline tests to explore the relationship between the key parameters of fluid and solid in the theoretical equation and their motion state. Durand [27] divided the flow state of solid–liquid two-phase flow into three categories based on the suspended flow state of sand and gravel in water and took the particle size as the defining standard. Fei [28] believed that, when the drag force of the water flow exceeds the starting drag force, the solid particles enter the state of motion, and their motion forms differ according to the intensity of the water flow. From the perspectives

of the supporting force of the motion of solid particles and the energy consumption of transportation, the motion forms of solid particles are divided into moving motion and suspended motion. Xu et al. [29] proposed the proportion coefficient K of solid particles in sliding and jumping motion to the total mass of solid particles and believed that when k = 1, the solid particles are completely in the moving state and move along the bottom of the pipe; when 0 < k < 1, some solid particles are in bed load state and some are in suspended load state; and when k = 0, the solid particles are completely in a suspended state. The test results of Vlasák et al. [30] demonstrated that part of the bed load will form a static and stable accumulation layer in the pipeline, and when the pipe flow velocity is large, the main movement form of the bed load is the jumping type. MatouŠek et al. [20,21] found that particles will show stratification in the pipeline (accumulation layer and transport layer), and the particle gradation will have an impact on sports energy consumption. Alihosseini et al. [31] demonstrated that the size of coarse particles can affect their critical velocity more than the pipe wall roughness, but the overall velocity of particles is greatly affected by the pipe wall roughness.

Based on the generalized model test and the diagram of particle motion state photograph taken by the high-speed camera PTV observation system under 75 working conditions, this study divided the particle motion into three transportation states: aggregation, jump, and suspension, and studied the influence of particle size, pipeline velocity, and sediment transport rate on the vertical distribution characteristics of the motion velocity of particle groups in the pipeline under the three transportation states.

*4.2. Pipeline Particle Velocity*

Because of the particularity and complexity of solid–liquid two-phase flow, there is no unified consensus on the calculation method of the vertical distribution of particle velocity in the pipeline. At present, the research has mainly focused on the influence of pipe shape, pipe diameter, particle size, particle-specific gravity, particle concentration, and other factors on the flow velocity in horizontal pipelines. Fei [32] advocated the use of "critical non-silting velocity", which refers to the velocity at which solid particles change from a suspended state to sliding or rolling on the bed surface; Ding [33] advocated "critical velocity of sedimentation", which mainly refers to the velocity from jumping sliding flow to fixed sediment bed flow. Wang [34] adopted the "non impact critical flow rate"; that is, the flow when the pipeline continues working in a stable fixed bed and the friction loss is not greater than the friction loss when the critical flow rate is not reached—this formulation is rarely used. Xia et al. [25] studied the distribution law of coarse particle concentration in clear water pipe flow and constructed the functional relationship between the relative position height of the maximum concentration point of the particle vertical line and the pipe diameter, particle size, average flow velocity and gravitational acceleration. The variation law and characteristics of the minimum transport velocity of long strip particles in vertical upwelling were studied, and the calculation formula of the minimum transport velocity of long strip particles was obtained [35,36].

Foreign experts and scholars have also researched the particle flow rate in the pipeline. Changhee et al. [37] studied the influence of the pipe shape on the flow rate and found that the flow rate in a square pipe is smaller than that in a circular pipe. Leporini [38,39] conducted numerical simulation research on sediment transport in multiphase fluid pipelines and studied an effective method to predict the velocity of sand particles by considering the influence of sand and gravel characteristics, sand concentration, flow pattern, fluid properties, pipeline properties, and other factors. Fajemidupe [40] studied the problem of sand deposition in oil pipelines, revealed the influence of the sand particle size and concentration on the flow resistance in gas–liquid stratified flow in horizontal pipelines, and proposed a prediction model for the minimum transportation conditions of sand transportation in horizontal pipelines. Dabirian [41,42] studied the critical deposition velocity of gas–liquid two-phase flow with air water glass marbles and found that when sand dunes were observed at the bottom of the pipeline, the critical deposition velocity increased slightly with

concentration, while for fixed beds, the critical deposition velocity increased exponentially with concentration. Januário et al. [43] calculated the critical deposition velocity of coarse particles in pipe flow. When the velocity is less than this, particles accumulate at the bottom of the pipe to form a static particle layer, and when the velocity is greater, particles no longer accumulate.

*4.3. The Scale Effect*

Scale effect that refers to a certain test law is only effective in a certain scale range; greater or less than this range, the law will be disturbed or even ineffective, so the "scale effect" must be considered in the design of the generalized model test. The generalization model test is based on the similarity of a certain shape size, mechanical index, and so on. To make the test results closer to the real or reflect the natural law, the optimal generalization scale is used to conduct the experimental research. For the pipeline flow generalizability test, Newit et al. [44] adopted a brass pipeline with a diameter of 2.54 cm (1 inch) to carry out the experimental model. The scale that is mainly focused on industrial application research is slightly larger; for example, the Salzgit Pipeline Experimental Center in Germany has a large pipeline system with a diameter of 50.8 cm (20 inches) [45]. In this study, the field test was used as the prototype to conduct the generalization test. The pipe diameter of the field test was 20 cm. From the perspectives of the test site, convenient observation, economy, similar shape, and similar mechanical indexes, a 10 cm pipe diameter was most appropriate for the generalization pipeline test. The Reynolds number of both the field test and the indoor generalization model was greater than $10^5$. The water flow state in the test was in the square resistance area, and the water flow motion law satisfied the physical equation of the square resistance area. In the analysis of test results, dimensionless relative quantities were used as variables of various empirical relationships in regression analysis, and the test results will have reference values for guiding field tests.

## 5. Conclusions

Based on the self-priming sand discharge of the reservoir pipeline, we studied the effects of particle size, pipeline velocity, and sediment transport rate on the vertical distribution characteristics of the movement velocity of particle groups in the pipeline under the three transport states of aggregation, saltation, and suspension through generalized model tests. This study provides a scientific and reasonable design basis for the distribution of sand discharge pipelines and is of great significance to further optimize the hydraulic transportation technology of pipelines. Under saltation transport, there is a critical inflection point between particle velocity and particle size in the vertical direction. Under aggregate transport and suspension transport, particle size is inversely proportional to particle velocity. In suspension transportation, the difference between the maximum and minimum vertical distribution velocities of particles has an exponential relationship with the pipeline velocity. Under the same pipeline velocity and particle size, the overall particle velocity decreases with the increase in sediment transport rate.

Because of limitations on the experimental conditions, the scaling model experiment for the self-priming pipeline discharge of a reservoir was not conducted, and only the generalization model was studied, which may have had scale effect problems. In addition, this study only simulated the motion law of coarse particles in the horizontal pipeline, and the motion characteristics of coarse particles in the anti-slope pipeline and the mutual feed mechanism of water flow were not involved, which will be the focus of the subsequent work of this study.

**Author Contributions:** Conceptualization, Z.W.; methodology, Z.L.; validation and formal analysis, L.Z.; investigation, resources, data curation, writing—original draft preparation, and writing—review and editing, J.W.; supervision, project administration, and funding acquisition, Y.L. All authors have read and agreed to the published version of the manuscript.

**Funding:** This research was funded by the Central Public-Interest Scientific Institution Basic Research Fund, grant number HKY-JBYW-2021-01; the National Natural Science Foundation of China, grant number 51909102; and the Key Commonwealth Project of Henan Province, grant number 201300311600.

**Institutional Review Board Statement:** Not applicable.

**Informed Consent Statement:** Not applicable.

**Data Availability Statement:** The data could be provided by the authors if necessary.

**Acknowledgments:** We would like to thank the potential reviewers very much for their valuable comments and suggestions. We also thank other colleagues for their valuable comments and suggestions that have helped improve the manuscript.

**Conflicts of Interest:** The authors declare no conflict of interest.

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
