# Peer review of "Study on the Motion Characteristics of Particles Transported by a Horizontal Pipeline in Heterogeneous Flow"

_water, doi:10.3390/w14193177_

Round 1

Reviewer 1 Report (Previous Reviewer 1)

The main improvements of this manuscripts have been made by the authors in previous revisions. In this sense, The manuscript can be published in the journal.  Some minor comments are:

1. References list does not have the reference style suggested by MDPI.

2. Table 1 should be improved the presentation.

3. Line 23 and 544.  Change “;” by “.”.

4. Line 138. Change “study” by “explore”.

5. In general, I recommend to use in a better way the template style of MDPI.

Author Response

1. References list does not have the reference style suggested by MDPI.

A:we had revised the references.

2. Table 1 should be improved the presentation.

A: we had revised the Table 1.

3. Line 23 and 544.  Change “;” by “.”.

A: we had revised them according to the comment.

4. Line 138. Change “study” by “explore”.

A: we had revised it.

5. In general, I recommend to use in a better way the template style of MDPI.

A: we had use the template style of MDPI.

Reviewer 2 Report (New Reviewer)

Please see the file attached.

Author Response

  1. However, I found very critical issues: (1) I don’t see in the manuscript any discussion on the scale effects and it appears unclear how the results could be scaled-up to the reality; (2)The proposed empirical formulas are provided in terms of dimensional variables and this would reduce the general impact of this study. But even worse, these formulas imply the sum of variables (e.g. the velocity and the sediment transport rate), which cannot be compared! Further specific comments are given below.

A:(1) In this study, the model experiment was started against the background of reservoir self-priming pipeline sand discharge, and the purpose of this study was to study the movement state of pipeline particles. In the early stage of the experiment, the scale effect was also planned to be studied, but due to the limitations of the experimental conditions, the scale modeling with the real hydraulic engineering was not carried out. Therefore, the main goal of this study is to study the movement state of pipeline particles through model tests, so as to provide a scientific basis for self-priming sand discharge in reservoirs.

  • The fitting of the formula in this study is still in the exploratory stage and may be immature. Therefore, we decided to delete all the fitting of the formula from the manuscript.
  1. The keywords “Physical model” and “Reservoir” appear too general and ineffective in

capturing the peculiarities of this study. I would substitute them for more specific ones.

A:We changed “Physical model” to “ Pipeline velocity”, we changed “Reservoir”to “Sediment transport rate”.

  1. [Horizontal pipeline particle conveying test device] (i) I have some perplexities on the quality of data collected in this study.  The pipe diameter was somewhat small (equal to 0.1 m) and the Reynolds numbers are not provided.  I’d like to see a subsection on the negligibility of scale effects;  (ii) How did the Authors assess the equilibrium conditions (in terms of sediment transport) along the horizontal effective observation section?  (iii) Were the experimental results dependent on the location of the “Dog-house” (see Figure 3)?

A: (1) In the real situation, the pipe diameter of the self-priming sand discharge of the reservoir pipeline is about 0.2m. The generalized model experiment was conducted in this study, so the pipe diameter of 0.1m was selected. The Reynolds number has been supplemented. As for the scale effect, this study does not consider it. This study aims to study the movement state of pipeline particles and its influencing factors by using generalization model experiment, and only takes the self-priming sand discharge of reservoir pipeline as the starting point.

    In this study, a generalized physical pipeline model with a fixed pipe length was adopted. The pipe diameter was 0.1m, the velocity of the secondary pipe in each experimental group was greater than 1m/s, the test temperature was 15°, the Reynolds number of all the groups was Re>105, and the water flow state in each group was in the square resistance area. For the flow state in the turbulent resistance square area of pipeline transportation are applicable. In the actual pipeline transportation of coarse sediment, the vast majority of water flow belongs to the turbulent resistance square area, so this test is representative, and some rules obtained from the test are not affected by the scale effect.

(2)The total length of the plexiglass pipe section is 23m. According to the numerical variation amplitude of four pressure sensors laid along the pipe, the middle 3m is determined as the effective observation section. Because the water flow in the pipe with the same diameter is uniform, the water flow conditions in each section of the effective observation section are the same, and the sediment transport conditions are also balanced.

(3)The total length of the plexiglass pipe section is 23m, and the feeding port is set 2m upstream of the horizontal effective observation section. The feeding port function is to add particles. For the pipeline water flow, the interference effect is small, and when the water flow reaches the observation section, the disturbance effect of feeding has already disappeared. (we revised Dog-house to Feeding port).

  1. [Results] (i) In the caption of Figure 9 the Authors write “Freudian number” instead of “Froude number”. Hopefully, this is a typo otherwise it would be a pretty serious mistake for a paper on hydraulic engineering topics! By the way, how did the Froude number is computed? (ii) The subsection 3.1 presents some empirical formulas that are absolutely unacceptable! For instance equation (3) is a dimensional equation and then remains unclear how it can be scaled-up to the reality. But even worse, it implies the sum of two variables - the velocity and the sediment transport rate - which cannot be compared!!!

A:(1)We are very sorry, this is a mistake, we had revised the “ Froude number”. We have supplemented the Froude number calculation method in the Methods section.

(2)(3)We had removed all the content of the fitting formula.

  1. Conclusions. This section is practically a reproduction of the Abstract; even the text is almost similar. In other terms this section should be rewritten. In particular, the Authors should: (i) not repeat what they have written in the Abstract; (ii) provide a synthesis of key-points; (iii) highlight the important features from this study; (iv) point out the limitations of this study; (v) indicate future directions.

A: We had revised the conclusions as follow:

Based on the self-priming sand discharge of reservoir pipeline, this study studied the effects of particle size, pipeline velocity and sediment transport rate on the vertical distribution characteristics of the movement velocity of particle groups in the pipeline under three transport states of aggregation, saltation and suspension through generalized model tests. This study provides a scientific and reasonable design basis for the distribution of sand discharge pipeline and is of great significance to further optimize the hydraulic transportation technology of pipeline. Under saltation transport, there is a critical inflection point between particle velocity and particle size in the vertical direction. Under aggregate transport and suspension transport, particle size is inversely proportional to particle velocity. In suspension transportation, the difference between the maximum and minimum vertical distribution velocity of particles has an exponential relationship with the pipeline velocity. Under the same pipeline velocity and particle size, the overall particle velocity decreases with the increase of sediment transport rate.

Limited by the experimental conditions, the scaling model experiment for the self-priming pipeline discharge of a reservoir was not carried out, and only the generalization model was studied, which may have some scale effect problems. In addition, this study only simulated the motion law of coarse particles in the horizontal pipeline, and the motion characteristics of coarse particles in the anti-slope pipeline and the mutual feed mechanism of water flow were not involved, which will be the focus of the subsequent work of this study.

6.[Figure 3] I would indicate the flow direction. Moreover, the text of the figure caption appears grammatically unstructured.

A:We had revised Fig. 3 as follow:

Fig. 3 Solid model generalization diagram

7.[Table 1] I would add a column with the values of the Reynolds number.

A: We had added the Reynolds number in Table 1.

Round 2

Reviewer 2 Report (New Reviewer)

To my concern "I don't see in the manuscript any discussion on the scale effects and it appears unclear how the results could be scaled-up to the reality" the Authors reply "...In the early stage of the experiment, the scale effect was also planned to be studied, but due to the limitations of the experimental conditions, the scale modeling with the real hydraulic engineering was not carried out....". I have appreciated the transparency of the Authors. However, I believe this is a crucial point - as remarked in my previous review. But I don't feel like to reject this article "tout-court". Therefore, I'd ask the Authors to add a subsection in the section "Discussion" dedicated to the possible scale effects that the experiments like those carried out by the them could involved. For the rest, revisions are acceptable though the manuscript would require a lot of refinements in terms of style and English language (especially in text marked in red). Among these I would point out the definition of the Froude number: equation (1) is not the Froude number, but its square! Moreover, Figure 4 is missing! 

In conclusion, my recommendation is "Reconsider after Major Revisions". A re-review is needed.  

Author Response

Thank you for your letter and for the reviewer’s comments concerning our manuscript. We have studied comments carefully and have made correction which we hope meet with approval. A version of the revised manuscript showing the new/changed text using track changes. The main corrections in the paper and the responds to the reviewer’s comments are described below.

We wish that the resubmit manuscript will meet the standard for publication. It would be greatly appreciated if <Water> could accept our paper for publication.

Thanks again for your time and letter. We are looking forward to hearing from you.

Yours sincerely,

Li Yitian

  1. To my concern "I don't see in the manuscript any discussion on the scale effects and it appears unclear how the results could be scaled-up to the reality" the Authors reply "...In the early stage of the experiment, the scale effect was also planned to be studied, but due to the limitations of the experimental conditions, the scale modeling with the real hydraulic engineering was not carried out....". I have appreciated the transparency of the Authors. However, I believe this is a crucial point - as remarked in my previous review. But I don't feel like to reject this article "tout-court". Therefore, I'd ask the Authors to add a subsection in the section "Discussion" dedicated to the possible scale effects that the experiments like those carried out by the them could involved.

A: We had added section 4.3 in the manuscript as follow:

4.3 The scale effect

Scale effect refers to a certain test law is only effective in a certain scale range, greater or less than this range, the law will be disturbed or even effective, so the "scale effect" must be considered in the design of generalized model test. The generalization model test is based on the similarity of a certain shape size, mechanical index and so on. In order to make the test results closer to the real or reflect the natural law, the optimal generalization scale is used to carry out the experimental research. For the pipeline flow generalizability test, Newit et al. [47] adopted a brass pipeline with a diameter of 2.54 cm (1 inch) to carry out the experimental model. The scale mainly focused on industrial application research is slightly larger, for example, Salzgit Pipeline Experimental Center in Germany has a large pipeline system with a diameter of 50.8cm (20 inches) [48]. In this study, the field test was used as the prototype to carry out the generalization test. The pipe diameter of the field test was 20 cm. From the perspective of the test site, convenient observation, economy, similar shape and similar mechanical indexes, 10cm pipe diameter was more appropriate for the generalization pipeline test. The Reynolds number of both the field test and the indoor generalization model is greater than 105. The water flow state in the test is in the square resistance area, and the water flow motion law satisfies the physical equation of the square resistance area. In the analysis of test results, dimensionless relative quantities are used as variables of various empirical relationships in regression analysis, and the test results have reference value for guiding field tests.

  1. For the rest, revisions are acceptable though the manuscript would require a lot of refinements in terms of style and English language (especially in text marked in red).

A: We had revised the language with MDPI.

  1. Among these I would point out the definition of the Froude number: equation (1) is not the Froude number, but its square!

A: We had revised equation (1) as follow:

The Froude number is calculated using the following equation:

                 Fr= (v2/gL)0.5                         (1)

Where, Fr, Froude number; v, the moving speed of the object; g, the acceleration of gravity; L, the characteristic length of the object.

  1. Moreover, Figure 4 is missing!

A: We had checked the manuscript, the Figure 4 does exist. We apologize for any inconvenience this may have caused you

This manuscript is a resubmission of an earlier submission. The following is a list of the peer review reports and author responses from that submission.

Round 1

Reviewer 1 Report

The current manuscript focused in the experimental study on the motion characteristics of particles transported in horizontal pipes.  The authors used an experimental facility to give an adequate interpretation of measured data. The authors compared experimental results versus empirical models. 

However, my major comment is related to the application of models based on physical equations. The manuscript does not bring relevant information about it. In this sense, I would like to see in the new version of the manuscript:         

-       A description of a model based on physical equations.

-       Comparison between a model based on physical equations versus experimental data.

-       Comments regarding of the application of the model in the Discussion section

My minor comments are:

-       The authors don’t use the complete template of MDPI. Please updated it. For instance, the bibliography section has double space.  Please use the template of 2021.

-       Tables and Figure don’t use the template.  The authors should correct all Tables. The enumeration of sub-figures is not presented.

-       In Figure 1 provides a photograph of the experimental facility. Also, the authors should include a photograph with relevant information where transport sediment can be detected in the horizontal pipeline.

-       The authors should be included a Figure that shows a description of the transport phenomena.

-       A notation section is missing in the manuscript.

Reviewer 2 Report

Comments on the manuscript: Study on the motion characteristics of particles transported by a horizontal pipeline in heterogeneous flow

The authors present experimental work carried out on measuring sand transport characteristics in a horizontal pipeline. After reading the title, I was very eager to read the manuscript and see how the work was carried out. However, it was rather disappointing as the most important part regarding the experimental methods and the results presentation was without sufficient detail and written like a magazine article not a research paper. Hence the reviewer cannot advise that the manuscript be accepted given the serious flaws in the measurement and a lack of information how they were obtained. Please refer to the comments below as my reason for this recommendation. Specifically:

  • fig. 2 is meaningless as since there is no annotation, it is not clear what it shows

  • Line 155: "An ultrahigh-speed intelligent particle size analyzer was used to analyze the particle size of the sediment samples" What is the model, what principle does it work on for measurement, what is the uncertainty in doing so, etc. All these were not given.

  • how did you measure the velocity profiles in figure 4? The procedure was not given in your experimental section. this should be described in detail.

  • following from above, the statements in your section 2.4 are unacceptable: "A symbolic regression method based on machine learning genetic coding was used to simulate the vertical motion velocity of pipeline particles, and the empirical formulas of vertical motion velocity of aggregation transportation, saltation transportation and suspension transportation of particles were obtained, providing a new method for studying the particle motion characteristics of sediment transportation via heterogeneous flow in a pipeline". They are too general and cannot be accepted for a scientific journal article, sounds like something you will write in a magazine or product advertisement article. One would expect each of the methods for velocity measurement, etc would be described with equations and uncertainties.

  • for example, PIV based methods can be used to determine the velocity field of the particles.
  • No sample images of the experiments are shown for different cases
  • the velocity profiles in fig. 4 and 5, 6a, and 7 are not correct. Did you change the axes by mistake? Please have a look at other papers or textbooks how velocity profiles should look like - for example the von Karman law of the wall. This relates to both the mixture velocity and particle velocities.

  • In addition all experimental measurements need to be accompanied with error bars indicating the level of uncertainty

  • fig. 3: the legends are rather confusing especially the blue one which should be the cumulative frequency or cumulative distribution of the particle sizes

  • some key works in particle transport such as the works of Leporini et al., Fajemidupe et al., Dabirian et al. have not been consulted and cited to support your method, hence the many mistakes that have been made in your measurements.
  • section 3.2 that derives correlations based on the doutbful "experimental" data cannot be trusted. Once the comments above have been addressed, then subsequent analysis can be credible.